# A Computational Approach to Explore the Interaction of Semisynthetic Nitrogenous Heterocyclic Compounds with the SARS-CoV-2 Main Protease

**DOI:** 10.3390/biom11010018

**Published:** 2020-12-27

**Authors:** Alejandro Llanes, Héctor Cruz, Viet D. Nguyen, Oleg V. Larionov, Patricia L. Fernández

**Affiliations:** 1Centro de Biología Celular y Molecular de Enfermedades, Instituto de Investigaciones Científicas y Servicios de Alta Tecnología (INDICASAT AIP), Ciudad del Saber, Panama 0801, Panama; allanes@indicasat.org.pa (A.L.); hcruz@indicasat.org.pa (H.C.); 2Department of Chemistry, University of Texas at San Antonio, One UTSA Circle, San Antonio, TX 78249, USA; vietnguyen2710@gmail.com

**Keywords:** SARS-CoV-2, main protease, semisynthetic compounds, molecular docking simulation

## Abstract

In the context of the ongoing coronavirus disease 2019 (COVID-19) pandemic, numerous attempts have been made to discover new potential antiviral molecules against its causative agent, SARS-CoV-2, many of which focus on its main protease (M^pro^). We hereby used two approaches based on molecular docking simulation to explore the interaction of four libraries of semisynthetic nitrogenous heterocyclic compounds with M^pro^. Libraries L1 and L2 contain 52 synthetic derivatives of the natural compound 2-propylquinoline, whereas libraries L3 and L4 contain 65 compounds synthesized using the natural compound physostigmine as a precursor. Validation through redocking suggested that the rigid receptor and flexible receptor approaches used for docking were suitable to model the interaction of this type of compounds with the target protein, although the flexible approach seemed to provide a more realistic representation of interactions within the active site. Using empirical energy score thresholds, we selected 58 compounds from the four libraries with the most favorable energy estimates. Globally, favorable estimates were obtained for molecules with two or more substituents, putatively accommodating in three or more subsites within the M^pro^ active site. Our results pave the way for further experimental evaluation of the selected compounds as potential antiviral agents against SARS-CoV-2.

## 1. Introduction

Since the beginning of the coronavirus disease 2019 (COVID-19) pandemic, declared on February 2020, many efforts have been made for the identification of potential antivirals against its causative agent, the severe acute respiratory syndrome (SARS)-related coronavirus called SARS-CoV-2. Several studies have focused mainly on impairing the interaction with the human angiotensin-converting enzyme 2 (ACE2) receptor or on inhibiting several viral proteins. The viral proteins mostly used as drugs targets include the RNA-dependent RNA polymerase (RdRp) and two proteases involved in cleavage of viral polyproteins, known as papain-like protease (PL^pro^) and 3-chymotrypsin-like protease (3CL^pro^) [1]. The latter, also known as the main protease (M^pro^), is essential for viral replication and is widely regarded as the most attractive drug target in SARS-related coronaviruses [2].

The native structure of M^pro^ is a dimer of two identical subunits, each one composed of a single polypeptide chain of ~300 amino acid residues [3,4,5]. Each subunit can be divided into three well defined domains (I, II and III), with the active site located between domains I and II. Domains II and III are connected by a relatively large linker loop that also flanks the active site. Two residues, His41 and Cys145, which form a catalytic dyad in the active site, primarily assist the proteolytic activity of M^pro^. The M^pro^ of SARS-related coronaviruses cleaves the viral polyproteins in several positions using core sequences to recognize the cleavage sites. Residues before and after the cleavage site in the substrate are named, respectively, P1 and P1′. Subsequent residues in both directions follow the same pattern in numbering, i.e., P2, P3, P4 and P5 indicate the four residues preceding P1 before the cleavage site. The active site of M^pro^ can accommodate at least six amino acid residues of the protein substrate, defining highly specific binding subsites often named S5 to S1′ after the corresponding residues.

Since the 2003 SARS outbreak, several molecules have been found to inhibit the M^pro^ of SARS-CoV and SARS-related coronaviruses (see [2] for a comprehensive review). Characterized inhibitors can be divided into generally covalent peptidomimetic inhibitors and small molecule inhibitors, the latter typically considered as non-covalent or reversible. Among the peptidomimetic inhibitors, the best studied is probably the N3 inhibitor, designed to target the SARS-CoV M^pro^ [6] and further found to inhibit the M^pro^ of other coronaviruses, including the infectious bronchitis virus (IBV) [4] and SARS-CoV-2 [5]. The N3 inhibitor has groups accommodating in subsites S5 to S1′ and forms a covalent bond with the Cys145 residue, all of which ensure a strong binding to the active site and a high inhibitory effect. Several studies have applied strategies based on drug repurposing [7,8,9,10,11,12] and the discovery of novel drug candidates [5,13,14,15,16,17,18] to identify inhibitors of M^pro^ as potential therapeutic agents against SARS-CoV-2. Although particular attention has been given to covalent peptide inhibitors such as N3, authors have also highlighted nitrogenous heterocyclic compounds as potential non-covalent inhibitors of the M^pro^ of SARS [2,19,20] and SARS-CoV-2 [21]. Based on these promising results, we hereby used a docking approach to computationally explore the interaction of four libraries of semisynthetic *N*-heterocyclic compounds with the SARS-CoV-2 M^pro^.

*N*-heterocyclic compounds are broadly represented among antiviral agents [22]. We therefore have constructed four libraries of *N*-heterocyclic compounds that were systematically selected after consideration of structural diversity, novelty, synthetic accessibility and druglikeness, the latter inferred on the basis of criteria such as compliance with the Lipinski’s rule of five (ROF score ≤ 1) [23]. Two broad classes of *N*-heterocycles were chosen based on reported antiviral activities for some representative members of each class. Quinolines have a long history as antiviral agents, and an assortment of structurally novel analogues of 2-alkylquinoline natural products represented by 2-propylquinoline (library L1) (Appendix A) were synthesized following a recently reported procedure [24]. A related subclass of compounds, quinoline *N*-oxides (library L2) (Appendix A), have recently emerged as promising and understudied derivatives of *N*-heterocycles with expanding applications in medicinal chemistry [25]. These compounds are readily synthetically accessible by oxidation of the parent *N*-heterocycles [26]. Libraries L3 and L4 (Appendix A) are based on hexahydropyrrolo[2,3-*b*]indole (HPI) natural products represented by physostigmine, which have been reported to exhibit an array of antiviral activities [27]. Library L3 comprises analogues with the truncated pyrrolidine ring, while library L4 contains structural analogues with the pyrrolidine ring bioisosterically replaced by the partially saturated 1,2-oxazine. Furthermore, 1,2-oxazine systems are encountered in several classes of biologically active natural products, most notably in the trichodermamide series [28,29]. These structurally novel derivatives of HPI natural products can be readily accessed by an inverse electron demand hetero-Diels-Alder reaction of indoles and transient nitrosoalkenes [30]. Our previous studies have indicated that the truncated and 1,2-oxazine-modified derivatives of HPI natural products possess divergent activities, suggesting that both types of analogues should be evaluated to gain insights into their structure–activity relationship [31,32].

## 2. Materials and Methods

### 2.1. Preparation of Receptor and Ligand Structures

We used the 1.83 Å crystal structure of SARS-CoV-2 M^pro^ with Protein Data Bank (PDB) identifier (ID) 5R84 (Fearon et al., unpublished) as the receptor for molecular docking simulations. The structure of the enzyme was prepared for docking by using the DockPrep module of Chimera v. 14 [33]. This step involved the removal of all crystallographic waters and ions, addition of missing hydrogens and side chains and assignment of appropriate charges and protonation states to ionizable amino acids at physiologic conditions (pH = 7.0).

Ligand structures were drawn with ACD/ChemSketch v.2020.1.1 (ACD/Labs, Toronto, ON, Canada) and the corresponding simplified molecular-input line-entry system (SMILES) notations were imported into Chimera for three-dimensional (3D) structure generation. Chimera was additionally used to prepare the molecules for docking with the DockPrep module, as described above for the receptor, and to optimize the geometry of the molecules through an energy minimization step before docking.

### 2.2. Molecular Docking Simulations

We used two approaches for molecular docking simulations, DOCK v. 6.9 [34], which was used to perform conventional flexible ligand docking, and Smina v. 2019-10-15 [35], a fork of AutoDock Vina v. 1.2.2 [36], which was used to perform a similar flexible ligand docking but with flexibility added to a subset of key amino acid residues in the active site of the receptor. The co-crystallized ligand in PDB 5R84 was used to define the position of the active site and to set box/grid parameters for both programs. Ten amino acid residues predicted to interact with this co-crystallized ligand were set as flexible when performing docking with Smina (using the --flexres option of the program), namely Phe140, Asn142, Gly143, Cys145, His164, Glu166, AsS187 and Gln189. Docking energy scores were calculated with the Hawkins Generalized Born/Surface Area (GBSA) scoring function for DOCK and with the custom scoring function for Smina.

## 3. Results and Discussion

### 3.1. Validation of Molecular Docking Parameters and Criteria for Selection

In the PDB structure used here as a receptor for molecular docking simulations, the SARS-CoV-2 M^pro^ was crystalized in complex with ligand 2-cyclohexyl-*N*-(3-pyridyl)acetamide, here called GWS according to its PDB ligand identifier. This molecule is a nitrogenous heterocyclic compound similar in size and chemical nature to many of the molecules considered in this study and therefore, it allowed us to define the position of the active site for docking (Figure 1A). Within the active site of M^pro^, GWS extends linearly across the S1 and S2 subsites and forms two hydrogen bonds with residues His163 and Glu166 (Figure 1B; Appendix A). The molecule also forms several non-covalent interactions with other residues from the active site, including His41, Met49, Phe140, Cys145, His164, Met165, AsS187 and Gln189, the latter two located in the linker loop of domains II and III.

In order to validate our docking parameters, we performed the redocking of the co-crystalized ligand with both docking programs before conducting the actual simulations with the target compounds. Poses with the lowest energy scores obtained through conventional rigid receptor docking with DOCK (–30.3 kcal/mol) and Smina (–6.3 kcal/mol) were relatively similar, with values of root mean square deviation (RMSD) of 2.137 and 1.22 Å, respectively. However, both poses were predicted to form only one of the two hydrogen bonds that GWS forms with residues from the active site. In an attempt to reproduce a more realistic scenario for the docking simulations, we reran Smina with similar parameters but adding flexibility to ten amino acid residues predicted to interact with GWS in the original PDB structure (see Section 2.2), which resulted in a lower, more favorable, energy estimate (–9.7 kcal/mol). The pose obtained with this flexible docking approach was more similar to the original orientation of the compound in the crystal (RMSD of 0.685 Å) and was predicted to form both hydrogen bonds with residues His163 and Glu166, as well as other non-covalent interactions with other residues in the active site (Figure 1; Appendix A). Since GWS is a nitrogenous heterocyclic compound, these results suggest that the docking parameters used in these redocking steps are suitable for our target compounds. For further analysis of molecular interaction of these target compounds and M^pro^, we prioritized the poses obtained with the flexible docking approach with Smina.

After performing docking simulations for the target compounds with the selected parameters, all molecules could fit within the M^pro^ active site, although individual poses varied significantly among members of the four libraries. Molecular docking simulations resulted in DOCK’s GBSA energy scores of –21.2 to –39.4 kcal/mol (Figure 2A) and Smina affinity scores of –6.7 to –13.4 kcal/mol (Figure 2B). Predicted energy estimates and the lowest energy poses for all the compounds and their natural precursors are shown in Appendix A. The difference in scale between these two types of energy scores is a result of the two programs using different algorithms and therefore these estimates cannot be directly compared. Although the word affinity is used in the definition of Vina and Smina energy scores, these values are not related to the affinity of the molecules to the receptor, such as the dissociation (*K*_d_) or inhibition (*K*_i_) constants. Taking into account the inherent differences between the two types of energy scores, we subsequently treated them independently and used the values of the lower quartiles as soft cut-offs to further select the compounds with the most favorable energy estimates (Figure 2C). These values should not be considered as thresholds to predict the binding of compounds to the target protein. These cut-off values were used to prioritize compounds for further analysis of their molecular interactions within the M^pro^ active site.

### 3.2. Quinoline and Quinoline N-Oxide Derivatives

We evaluated 52 compounds belonging to libraries L1 and L2 including quinoline (compounds 1–25) and quinoline *N*-oxide (compounds 26–52) derivatives, respectively. Molecular docking simulations resulted in 11 compounds from each of the libraries L1 (Figure 3A) and L2 (Figure 3B) meeting the criteria defined for selection. Energy scores computed for the 2-propylquinoline natural precursor, –8.5 kcal/mol (DOCK) and –25.8 kcal/mol (Smina), were less favorable than those estimated for the vast majority of compounds in these two libraries. In the binding poses predicted for this precursor, which were similar for both programs, the molecule was oriented so that the quinoline nucleus accommodates between the S3 and S4 subsites and the propyl chain protrudes towards the S2 site, leaving the S1 and S1′ subsites empty. Binding poses for compounds 1 and 2 were similar to that of the precursor, but the presence of an additional phenyl group in compounds 3–7 appears to promote a more favorable orientation of the quinoline nucleus in the active site, with the best energy scores estimated for compounds 4, 5 and 6. A similar orientation was predicted for compounds 9 and 15, which have an additional cyclopentane ring at position C2 of the quinoline nucleus with one or two additional Cl substituents; as well as for compound 17, which is similar to compound 15 but has a 4-chlorophenyl group at C2 instead of the cyclopentane ring.

Compounds 20 and 21 have a 4-fluorophenyl group linked through an ether bond to C4 that accommodates around the S1 and S2 subsites, possibly contributing to more favorable energy estimates, but still leaving the area around the catalytic dyad and the S1′ subsite empty. Similar orientations were predicted for compounds 23, 24 and 25, which also have relatively large substituents attached to C4 through a secondary amine bond and containing a tertiary amine moiety. The only compound of library L1 that was predicted to fill the active site in a way resembling that of inhibitor N3 was compound 22, which is also the only compound in the library with two stereoisomers (designated 22_R and 22_S). Although energy scores estimated for this compound were not the lowest in the library, this is the most attractive candidate for experimental evaluation among the quinoline derivatives, since both stereoisomers were concurrently predicted to fill all the five subsites in the active site (Figure 4). Both stereoisomers were predicted to form a hydrogen bond with His164, a residue that is sterically close to the His41 and Cys145 residues forming the catalytic dyad. The 22_R stereoisomer was also predicted to form an additional hydrogen bond with residue Asn142. Compounds 22–25 were constructed to mimic antimalarial drugs chloroquine (22) and amodiaquine (compounds 23–25), both of which were recently shown to have notable antiviral activities against SARS-CoV-2 [37,38,39]. However, clinical effectiveness of hydroxychloroquine in the treatment of COVID-19 patients has been recently questioned [40].

Globally, quinoline *N*-oxide derivatives from library L2 received relatively lower energy scores when compared to the quinoline derivatives in library L1. These compounds tended to accommodate in the relatively large cavity formed in the active site around subsites S2, S3, S4 and S5, in some cases forming a hydrogen bond between the charged oxygen atom at position N1 of the quinoline nucleus and the Gln189 residue in the linker loop. Of the compounds selected from this family, only compounds 48, 49, 50 and 52, containing trifluoromethyl groups as substituents, were concurrently predicted by both programs to cover a wider region of the active site, involving the S1 and S1′ subsites. Docking poses for compounds 48 and 49 are very similar to the orientation of GWS in the crystal, with the molecules extending almost linearly along the S1 and S2 subsites and, in the case of compound 48, forming a hydrogen bond with residue Glu166. In compounds 50 and 52, the relatively bulky phenyl ring with two trifluoromethyl substituents was predicted to accommodate around the S1′ and S4 subsites, respectively, causing the quinoline nucleus to position very close to residue Cys145.

### 3.3. Derivatives of Hexahydropyrrolo[2,3-b]indole (HPI) Natural Products

We also evaluated 65 compounds from two additional libraries of HPI derivatives, L3 and L4, containing the truncated pyrrolidine ring (compounds 53–90) and the tetrahydro-1,2-oxazino indole (TOI) nucleus (compounds 91–117), respectively. Molecular docking simulations resulted in 14 compounds from library L3 (Figure 5) and 22 compounds from library L4 (Figure 6) meeting the criteria for selection. The energy score estimated with DOCK for physostigmine (–26.5 kcal/mol), which is the natural precursor of these libraries, was less favorable than those computed for most of the compounds in both libraries. However, the Smina score computed for this precursor (–10.1 kcal/mol) was more favorable than that computed by DOCK, when referring to the individual thresholds selected for both programs. In the lowest-energy poses predicted by these two programs, which were also roughly consistent, the molecule extends across the S1 and S2 sites.

Binding poses predicted for several compounds belonging to library L3 followed the same orientation predicted for physostigmine, although the presence of one or two additional hydrophobic and bulky substituents at positions N1 or C3 of the indole nucleus typically resulted in lower, more favorable, energy estimates. For instance, compounds 53 and 69, which, respectively, have a methyl and an isopropyl group in position N1, received relatively unfavorable scores. However, compound 63, which differs from compound 53 in the presence of a 2,5-dimethylbenzyl group in N1, received more favorable energy scores. In the binding poses for most of these compounds, the additional bulky substituents appear to accommodate around the S2/S4 subsites or the S1/S1′ subsites. No molecule from this library was concurrently predicted by both programs to occupy more than three subsites in the active site. Additionally, due to the general lack of donor or acceptor groups in compounds from this library, almost none of them were predicted to form hydrogen bonds with residues from the active site. However, some of the energy scores computed for these molecules were similar or better than those estimated for compounds from the other libraries forming one or more hydrogen bonds. This suggests that other non-covalent interactions, including hydrophobic interactions, may have the largest contribution to energy scores estimated for compounds from this library, particularly in those with highly hydrophobic or aromatic substituents. Compound 64 was the only molecule in the library that was predicted to form a hydrogen bond with a residue from the active site (Gln189), in this case due to the presence of a carbonyl group at C3.

On the other hand, the compounds containing the tetrahydro-1,2-oxazino indole (TOI) nucleus from library L4 received the most favorable energy scores among all the four libraries examined here. Unlike those from all other libraries, compounds from this library have four stereoisomers due to the presence of two stereocenters, located at positions 4a and 9a of the TOI nucleus. Here, the four stereoisomers were designated with suffixes “**_**RR” (4a*R*,9a*R*), “**_**RS” (4a*R*,9a*S*), “**_**SR” (4a*S*,9a*R*) and “**_**SS” (4a*S*,9a*S*). Each of these stereoisomers was submitted separately to the docking simulations. Poses predicted for the four stereoisomers were notably different for almost all compounds in the library. However, stereoisomers with the most favorable energy scores in the library, including 97_RR, 104_RR, and 110_SR, tended to adopt a similar orientation within the active site, despite having substituents of different chemical nature (Figure 7). In this conserved orientation, the TOI nucleus occupied a relatively central position in the active site, very close the catalytic dyad, while the substituents in positions 3, 4a and 9 accommodate in the S4, S2 and S1 subsites, respectively. This orientation within the active site also appears to be favored by the formation of a hydrogen bond between the oxygen atom of the oxazine ring of the TOI nucleus and residue Glu166. All these findings suggest that these three stereoisomers and/or the corresponding racemates are the most attractive candidates for experimental evaluation among the compounds in this library.

Several other stereoisomers with favorable energy scores followed a similar orientation, with relatively small changes in the local position of the TOI nucleus and its substituents but occupying the same subsites. These include 105_RR, 106_RS, 107_RR, 108_RR, 111_SR, 112_SR, 116_SR and three stereoisomers of compound 117 (117_RR, 117_RS and 117_SR). Finding regularities that can serve as hypotheses about binding of these compounds to the active site is difficult due to the diverse nature of their substituents, but the presence of a hydrophobic or bulky group at position C3 again seems to be relevant. The Br substituent at position C6 also appears to favor positioning of the TOI nucleus, since in almost all the compounds with this substitution there is at least one stereoisomer in which the halogen atom orients towards the S1′ subsite. Conversely, in all the stereoisomers of compound 116, which lack this substituent at C6 but have a 4-bromophenyl substituent at C3, the TOI ring accommodates so that the 4-bromophenyl group is oriented towards S1′. In compound 117, which have an I instead of a Br at position C6, the three stereoisomers mentioned above have the halogen atom oriented towards S1′.

## 4. Conclusions

This study proposes a methodology for rapid prediction of binding of *N*-heterocycles to the M^pro^ of SARS-related coronaviruses. We used two molecular docking simulation approaches to explore the interaction of four libraries of *N*-heterocyclic compounds with the SARS-CoV-2 M^pro^. Globally, we observed no significant correlation between the results obtained through the rigid receptor approach (DOCK) and the flexible receptor approach (Smina), although both approaches tended to agree better in molecules with favorable energy estimates. Docking of co-crystalized ligand and careful inspection of molecular interactions of target compounds suggested that the flexible approach, which adds flexibility to 10 key amino acid residues of the M^pro^ active site, was a better model to predict the orientations and interactions within the active site. Using empirical energy score thresholds, we selected 58 compounds from the four libraries to further discuss their predicted interaction with M^pro^. Most favorable predictions were obtained for relatively large molecules from libraries L1 and L4, containing substituents capable of accommodating in three or more subsites within the M^pro^ active site. The presence of two or more large, relatively bulky substituents is common in previously reported inhibitors of the SARS-CoV M^pro^, including peptidomimetic inhibitors containing Michael acceptors and other non-peptidic molecules [2].

Due to the relevance of M^pro^ in viral replication, this protein has been considered one of the most attractive targets for the identification of new antiviral drugs against SARS-related coronaviruses, including SARS-CoV-2 [1,2]. However, despite all efforts thus far, no M^pro^ inhibitor has advanced to clinical trials. The identification of new molecules with the potential to inhibit viral replication and consequently the infection, continues to be urgent. Herein, we have identified several compounds predicted to interact with M^pro^. Further studies are necessary to evaluate, in vitro, the effect of those compounds on SARS-CoV-2 replication. The potential advance of new molecules in preclinical and clinical programs offers hope for the development of effective therapies that help control the spread of the virus. This work adds to the efforts of the scientific community in the fight against SARS-CoV-2.

## Figures and Tables

**Figure 1 biomolecules-11-00018-f001:**
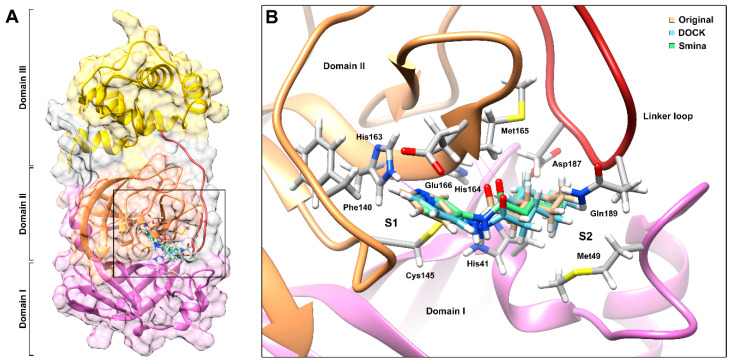
Redocking of the ligand co-crystalized with chain A of SARS-CoV-2 main protease (M^pro^) in Protein Data Bank (PDB) 5R84. (**A**) Location of the active site relative to the domains I, II and III of M^pro^, respectively, colored purple, orange and yellow. Linker loop connecting domains II and III is colored dark red. (**B**) Comparison of docking poses obtained with DOCK and Smina (flexible docking approach) with the original orientation of the co-crystalized ligand. Hydrogen bonds with residues His163 and Glu166 are represented with thin purple lines.

**Figure 2 biomolecules-11-00018-f002:**
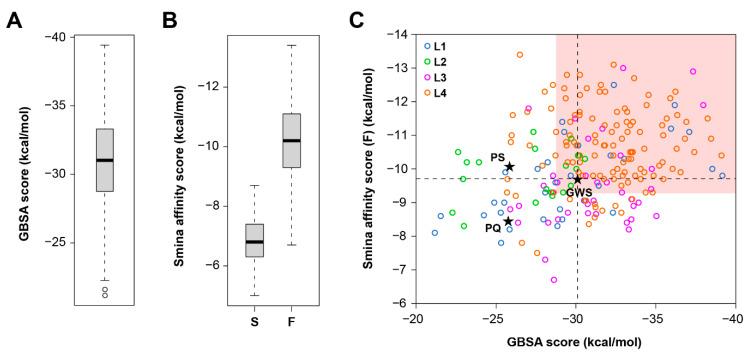
Values of docking energy estimates obtained for the target compounds. (**A**) Box-and-whisker plot for the DOCK energy scores. (**B**) Box-and-whisker plots for the Smina affinity scores obtained with standard (S) rigid receptor docking and the flexible receptor (F) approach. (**C**) Distribution of values among the four libraries of compounds evaluated (L1, L2, L3 and L4), with compounds meeting the selection criteria clustering in the area highlighted in pink. Energy scores obtained for the co-crystalized ligand (GWS) are indicated with dashed lines. PQ: 2-propylquinoline, PS: physostigmine.

**Figure 3 biomolecules-11-00018-f003:**
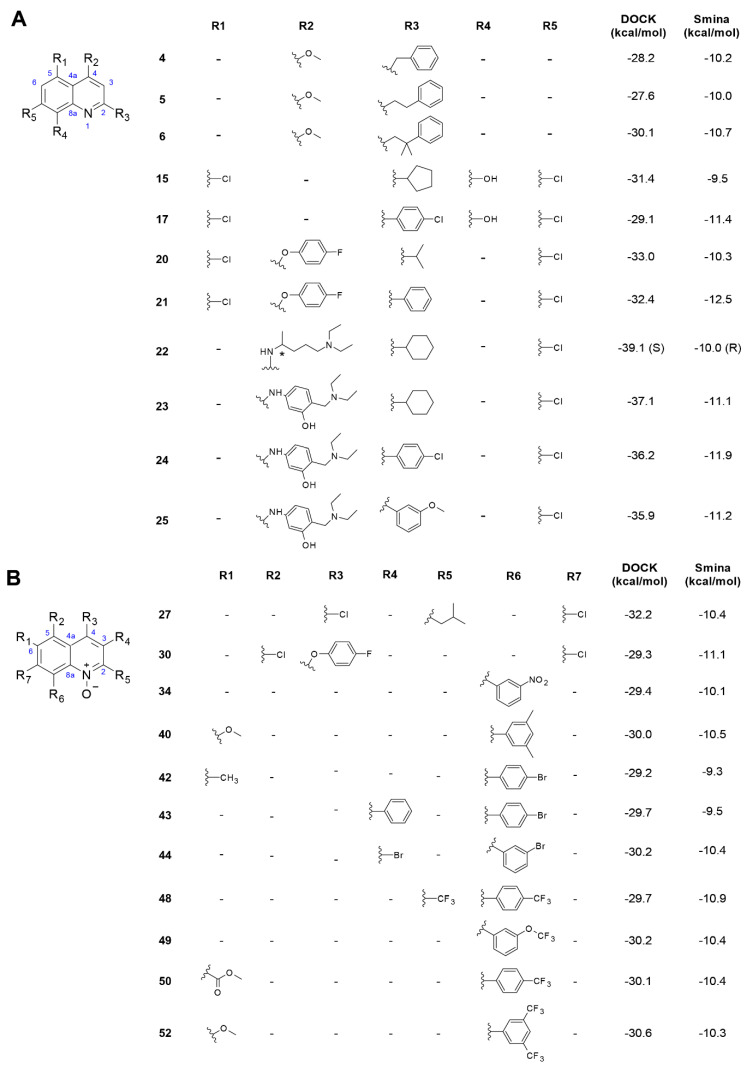
Compounds from libraries L1 (**A**) and L2 (**B**) with the most favorable energy scores. The compounds were selected according to the cut-off values defined in Section 3.1. For compound 22, position of the stereocenter is indicated with “*” and the most favorable, lowest, energy score between the two stereoisomers is shown for each program.

**Figure 4 biomolecules-11-00018-f004:**
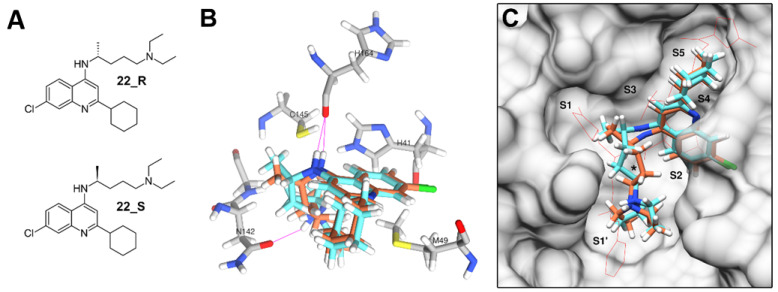
Docking poses predicted for compound 22. (**A**) Structure of stereoisomers 22_R and 22_S. (**B**) Orientation predicted for stereoisomers 22_R (cyan) and 22_S (orange) in the active site of M^pro^. Hydrogen bonds are represented by violet lines. (**C**) Extent of the orientation predicted for both stereoisomers in the active site, compared with the orientation of the N3 inhibitor (red wire). Approximate position of the His41-Cys145 catalytic dyad is indicated with “*”.

**Figure 5 biomolecules-11-00018-f005:**
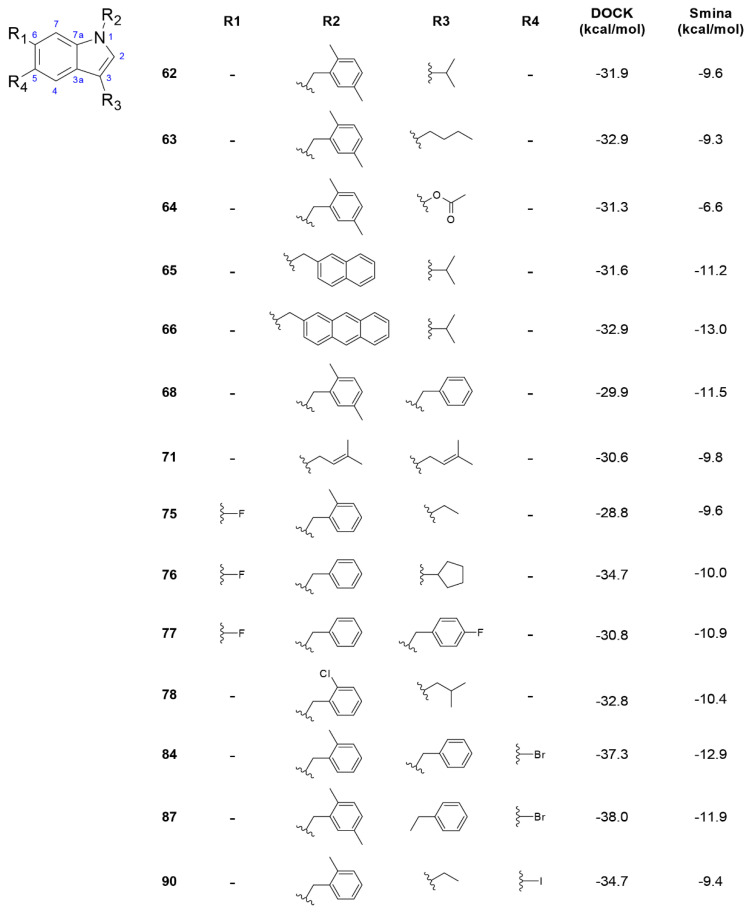
Compounds from L3 library with the most favorable energy scores. The compounds were selected according to the cut-off values defined in Section 3.1.

**Figure 6 biomolecules-11-00018-f006:**
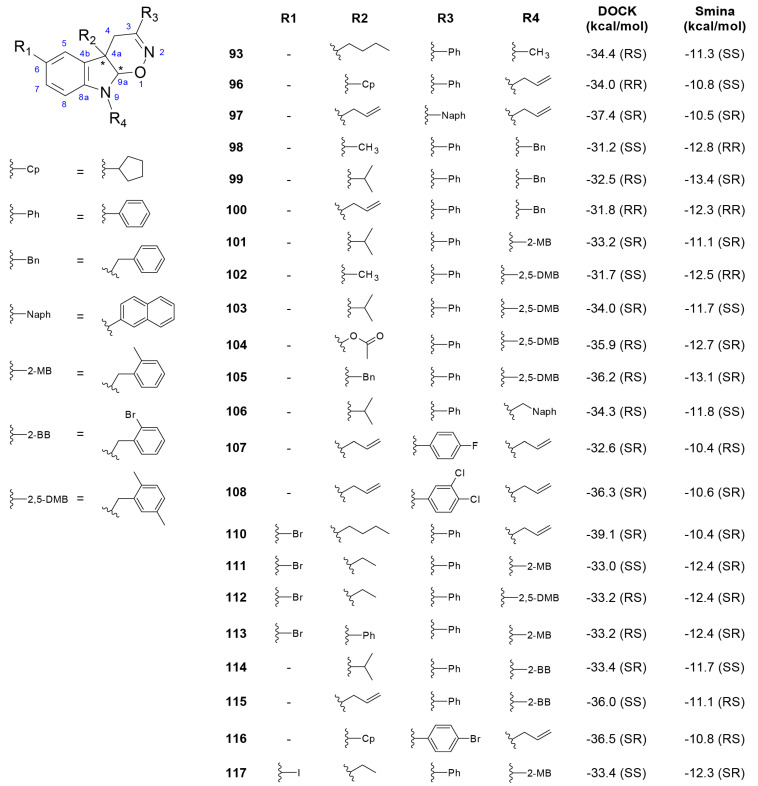
Compounds from L4 library with the most favorable energy scores. The compounds were selected according to the cut-off values defined in Section 3.1. All the selected compounds have at least one stereoisomer meeting the selection criteria. Position of stereocenters in the tetrahydro-1,2-oxazino indole (TOI) nucleus is indicated with “*”. Only the value for the stereoisomer with the most favorable, lowest, energy score is shown for each compound and program. Cp: cyclopentyl; Ph: phenyl; Bn: benzyl; Naph: naphthyl; 2-MB: 2-methylbenzyl; 2-BB: 2-bromobenzyl; 2,5-DMP: 2,5-dimethylphenyl.

**Figure 7 biomolecules-11-00018-f007:**
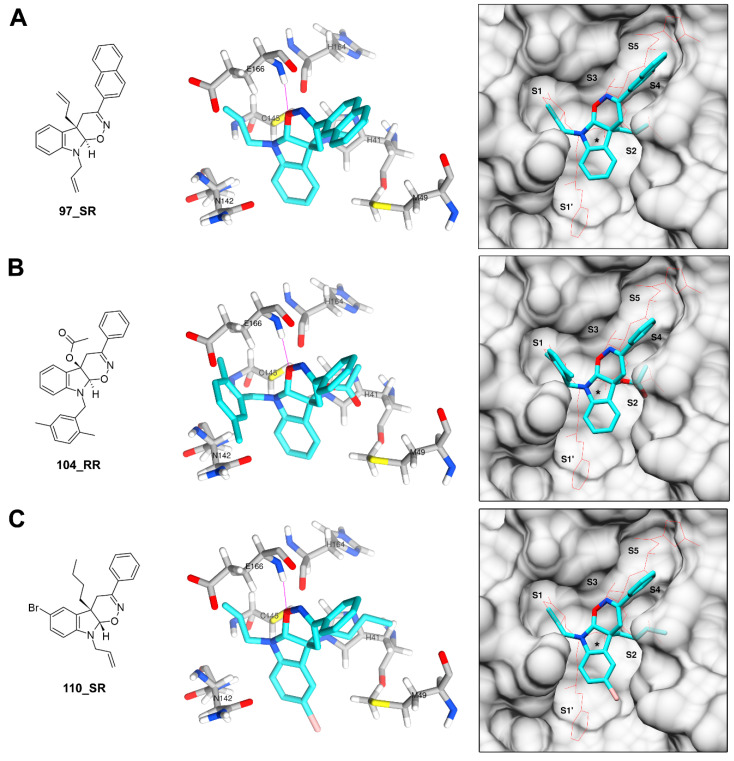
Docking poses predicted for stereoisomers 97_RR (**A**), 104_RR (**B**) and 110_SR (**C**). These stereoisomers are colored in cyan with hydrogen bonds represented by violet lines. The orientation of the N3 inhibitor in surface representations is indicated by a red wire and the approximate position of the His41-Cys145 catalytic dyad is indicated with “*”.

## Data Availability

Data is contained within the article or Appendix A.

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
