# Peer review of "A Computational Approach to Explore the Interaction of Semisynthetic Nitrogenous Heterocyclic Compounds with the SARS-CoV-2 Main Protease"

_biomolecules, 2020, doi:10.3390/biom11010018_

Round 1

Reviewer 1 Report

The manuscript titled, "A computational approach to explore the interaction of semi-synthetic nitrogenous heterocyclic compounds with the SARS-CoV-2 main protease ", is well written and may be accepted for publication.

The idea of doing computational study using one or more computer aided programs is not new, but at this time helping to build more library content. It was not surprising to see that the analogs that were constructed to mimic antimalarial drugs chloroquine had the best hit. It would have been interesting to see new molecules or different class of molecules created by either linkers or just pure chemical modifications.

Author Response

Reviewer 1

The manuscript titled, "A computational approach to explore the interaction of semi-synthetic nitrogenous heterocyclic compounds with the SARS-CoV-2 main protease ", is well written and may be accepted for publication.

The idea of doing computational study using one or more computer aided programs is not new, but at this time helping to build more library content. It was not surprising to see that the analogs that were constructed to mimic antimalarial drugs chloroquine had the best hit. It would have been interesting to see new molecules or different class of molecules created by either linkers or just pure chemical modifications.

We would like to thank the reviewer for the revision of the manuscript.

We tested four compounds that were constructed to mimic antimalarial drugs chloroquine and amodiaquine (compounds 22-25). The rest of the evaluated compounds are novel derivatives of the natural products 2-propylquinoline and physostigmine that are derived by introducing chemical modifications through recently described procedures.

Reviewer 2 Report

In the present Manuscript, the Authors optimize chemical compounds with potential binding activity to SARS-CoV-2 main protease and find different candidates that bind the active site of the protease. The Authors however do not provide any experiments, which would prove the activity of the compounds, and do not discuss their results with available literature. 

I have these major points that should be answered, if the article is accepted for publication:

i) In the Discussion the Authors should discuss applicability of their results in COVID-19 control.

ii) In the Discussion the Authors state the similarity of their best compounds to antimalarial and antileishmanial drugs. First, that is largely irrelevant, as malaria and leishmania are not viral diseases. Second, hydroxychloroquine have been shown by WHO to be ineffective https://doi.org/10.1101/2020.10.15.20209817 in COVID-19 treatment. The Authors should improve their Discussion.

iv) Most favorable predictions were obtained for large molecules. The Authors should at least discuss, if the compounds do or do not satisfy Lipinski's rule of five for orally administered drugs.

Author Response

In the present Manuscript, the Authors optimize chemical compounds with potential binding activity to SARS-CoV-2 main protease and find different candidates that bind the active site of the protease. The Authors however do not provide any experiments, which would prove the activity of the compounds, and do not discuss their results with available literature.

I have these major points that should be answered, if the article is accepted for publication:

i) In the Discussion the Authors should discuss applicability of their results in COVID-19 control.

We would like to thank the reviewer for the revision of the manuscript.

As suggested by the reviewer, we have added a paragraph explaining the relevance of the work at the end of the conclusions section.

ii) In the Discussion the Authors state the similarity of their best compounds to antimalarial and antileishmanial drugs. First, that is largely irrelevant, as malaria and leishmania are not viral diseases. Second, hydroxychloroquine have been shown by WHO to be ineffective https://doi.org/10.1101/2020.10.15.20209817 in COVID-19 treatment. The Authors should improve their Discussion.

From the four libraries evaluated only four compounds were synthesized to mimic antimalarial drugs chloroquine and amodiaquine (compounds 22-25). Although those compounds were among the best ranked in our study, several derivatives from library L4 were also very attractive candidates for experimental evaluation. We agree with the reviewer that similarity of compounds 22-25 to hydroxylchloroquine should not be used as a notion to prioritize these candidates for future experimental evaluation. Thus, we have removed all references to the analogy to antimalarials throughout the manuscript except from Discussion, when we included the following statement “Compounds 22-25 were constructed to mimic antimalarial drugs chloroquine (22) and amodiaquine (compounds 23-25), both of which were recently shown to have notable antiviral activities against SARS-CoV-2 [36–38]. However, clinical effectiveness of hydroxychloroquine in the treatment of COVID-19 patients has been recently questioned [39].” However, we believe that the lack of clinical effectiveness of chloroquine to treat COVID-19 patients should not be considered as a negative criterion against future evaluation of our compounds, since they do not have exactly the same chemical structure and a single chemical modification can have a great impact on the biological activity of organic molecules.

We mentioned the previously reported antileishmanial activity for compound 22 not because it is related to its possible antiviral effect, but to emphasize this compound has been previously evaluated in in vitro and in vivo assays, where it was found to be active and safe. However, we have removed from the manuscript the reference to antileishmanial activity of compound 22.    

iv) Most favorable predictions were obtained for large molecules. The Authors should at least discuss, if the compounds do or do not satisfy Lipinski's rule of five for orally administered drugs.

In the Introduction, we mentioned that compounds were selected following different criteria including druglikeness. The Lipinski's rule of five was taken into account to infer druglikeness and in order to clarify, the sentence now states “We therefore have constructed four libraries of N-heterocyclic compounds that were systematically selected after consideration of structural diversity, novelty, synthetic accessibility and druglikeness, the latter inferred on the basis of criteria such as compliance with the Lipinski's rule of five (ROF score ≤ 1).”

Reviewer 3 Report

The manuscript "A computational approach to explore the interaction of semisynthetic nitrogenous heterocyclic compounds with the SARS-CoV-2 main protease" is quite interesting however some points required more attention

Among these points:

1- The reason for selecting the main protease and not other protease like papain-like protease  and 3-chymotrypsin-like protease at least the active compounds should be also tested 

2- The location of the active sites of protein are not described in the manuscript or supplementary. The usage of a pH-based  method in this study is not described well

3-The selection criteria for these libraries are not clear 

4- Is there any possibility to measure the active compounds in vitro?

Author Response

The manuscript "A computational approach to explore the interaction of semisynthetic nitrogenous heterocyclic compounds with the SARS-CoV-2 main protease" is quite interesting however some points required more attention

Among these points:

1- The reason for selecting the main protease and not other protease like papain-like protease and 3-chymotrypsin-like protease at least the active compounds should be also tested

We would like to thank the reviewer for the revision of the manuscript.

Main protease and 3-chymotrypsin-like protease are alternative names for the same protein. We choose this protein as target because several of its experimentally validated inhibitors are N-heterocycles. We indeed considered evaluating the compounds with the papain-like protease (PLpro); however, potential side effects and cytotoxicity of such inhibitors resulting from the fact that the PLpro of SARS-related coronaviruses is similar to human deubiquitinating enzymes have recently been discussed in the literature (https://doi.org/10.1016/j.antiviral.2017.11.001; https://doi.org/10.3390/ph13100277). Despite these limitations, certain potential inhibitors of SARS-related coronaviruses have been reported (https://doi.org/10.1073/pnas.0805240105), but they are typically not N-heterocyclic in nature and many of them had to be subsequently modified to avoid improper biding to the active site. For these reasons, we decided to focus on Mpro.

2- The location of the active sites of protein are not described in the manuscript or supplementary. The usage of a pH-based method in this study is not described well

The location of the active site and the main amino acid side chains within it are shown in Figure 1. We used the ligand crystallized in complex with the main protease in our template PDB (5R84) to define the position of the active site for docking. We clarify this in paragraph 1 from section 3.1.

In the Materials and Methods section, the phrase “… assignment of appropriate charges and protonation states to ionizable amino acids at pH 7.0” refers to properly setting amino acid with ionizable groups at physiological or cellular conditions before conducting the docking per se. We rephrase the sentence as follows “… assignment of appropriate charges and protonation states to ionizable amino acids at physiologic conditions (pH = 7.0).”

3-The selection criteria for these libraries are not clear

As stated in Introduction, these compounds were selected among several molecules recently synthesized by our group based on previous reports of N-heterocycles as potential inhibitors of the main protease of SARS-related coronaviruses as well as other relevant features such as structural diversity, novelty, synthetic accessibility and druglikeness.

4- Is there any possibility to measure the active compounds in vitro?

Considering the scope of the special issue, the main goal for this work was to standardize a methodology for rapid evaluation of the interaction of semi-synthetic N-heterocycles with Mpro. Further studies will evaluate the effect of compounds on SARS-CoV-2 pseudoviruses.

Round 2

Reviewer 2 Report

The authors responded and clarified all the required points. The manuscript can be accepted in the present form.

Reviewer 3 Report

The authors responded and clarified all the required points

it could be accepted in the present form